# Identification and Diagnosis of Complete Haptoglobin Gene Deletion, One of the Genes Responsible for Adverse Posttransfusion Reactions

**DOI:** 10.3390/biomedicines12040790

**Published:** 2024-04-03

**Authors:** Mikiko Soejima, Yoshiro Koda

**Affiliations:** Department of Forensic Medicine, Kurume University School of Medicine, Kurume 830-0011, Japan; misoe@med.kurume-u.ac.jp

**Keywords:** adverse reactions, haptoglobin deficiency, haptoglobin gene deletion, serum protein, transfusion

## Abstract

Allergic reactions are the most frequent adverse events in blood transfusion, and anaphylactic shock, although less frequent, is systemic and serious. The cause of allergic reactions to blood transfusions are largely unknown, but deficiencies in serum proteins such as haptoglobin (Hp) can lead to anaphylactic shock. A complete deletion of the haptoglobin gene (*HP^del^*) was first identified in families with anomalous inheritance and then verified as a genetic variant that can cause anaphylactic shock because homozygotes for *HP^del^* have complete Hp deficiency. Thereby, they may produce antibodies against Hp from blood transfusions. *HP^del^* is found in East and Southeast Asian populations, with a frequency of approximately 0.9% to 4%, but not in other populations. Diagnosis of Hp deficiency due to *HP^del^* prior to transfusion is advisable because severe adverse reactions can be prevented by washing the red blood cells and/or platelets with saline or by administering plasma products obtained from an Hp-deficient donor pool. This review outlines the background of the identification of *HP^del^* and several genetic and immunological methods developed for diagnosing Hp deficiency caused by *HP^del^*.

## 1. Introduction

Blood transfusion and the administration of blood products are relatively safe medical procedures, but they carry the risk of adverse reactions [1]. Therefore, investigation of the causes of transfusion-related adverse reactions and their prevention is an important issue for transfusion medicine.

Adverse reactions to blood transfusion are mainly classified as adverse hemolytic reactions, adverse non-hemolytic reactions, transfusion-associated graft versus host disease, and infections [2]. In Japan, the frequency of adverse non-hemolytic reactions is the highest, with 2532 cases reported in a survey by the Japanese Red Cross Society in 2020, accounting for 96.1% of 2634 cases of all blood transfusion adverse reactions (Haemovigilance by JRCS 2020, “https://www.jrc.or.jp/mr/relate/info/pdf/Haemovigilance%20by%20JRCS%202020_JP.pdf” (accessed on 1 March 2024)). Among non-hemolytic adverse reactions, allergies were the most frequently reported, at 1692 cases, accounting for approximately two-thirds of all reported adverse reactions; 332 cases of which were severe allergic reactions such as anaphylaxis, which can be life-threatening. The causes of allergic reactions to blood transfusions are largely unknown. However, the causes that have been identified include deficiencies of plasma proteins such as immunoglobulin (Ig) A, haptoglobin (Hp), and complement component 9 (C9). Patients with these deficiencies have received blood transfusions in the past, causing severe allergic reactions due to the production of antibodies to these proteins [3,4,5,6,7,8,9]. It is known that IgA deficiency is more common in Europe and the United States, whereas Hp deficiency is more common than IgA and C9 deficiency in Japan [10,11]. In order to prevent severe allergic reactions due to transfusions of red blood cell and platelet products, transfusions of products from which plasma and other components have been removed by washing at least two times with saline at 4 °C (e.g., washed red blood cell products) have been reported to be effective [12]. In addition, when plasma products are transfused into a patient whose allergy is identified as being caused by a plasma protein deficiency, the derived plasma products from the donor should be deficient of the protein concerned [13]. Therefore, the diagnosis of plasma protein deficiency prior to transfusion is very important for safe blood transfusion.

In this review, we will outline the background of identification of complete haptoglobin gene deletion (*HP^del^*), its geographical distribution, and several methods to diagnose congenital Hp deficiency (congenital anhaptoglobinemia) due to *HP^del^* [14,15].

## 2. Characteristics and Polymorphism of Haptoglobin

Hp is an acute phase serum protein synthesized primarily in the liver [16]. It binds to the highly toxic free hemoglobin (Hb) to form a stable Hp–Hb complex [17]. This complex is rapidly cleared by binding to the macrophage scavenger receptor CD163 and being digested in lysosomes to release heme, thereby preventing Hb-induced oxidative stress [18]. Because Hp is one of the positive acute phase reactants, its serum concentration increases in various clinical conditions such as infection and inflammation [19]. On the other hand, during hemolysis, the Hp concentration decreases dramatically due to the rapid clearance of the Hp–Hb complex [20]. In addition, Hp can bind apolipoprotein A1 and E, which may affect lipid metabolism through its ability to reverse transport of cholesterol from peripheral cells to the liver [21,22,23].

As shown in Figure 1, the haptoglobin gene (*HP*) has an extra copy called the haptoglobin-related protein gene (*HPR*). All figures in this review (Figure 1, Figure 2, Figure 3 and Figure 4) have not been previously published. These two genes share a high degree of nucleotide sequence similarity, and *HPR* is located on chromosome 16, 2.2 kb downstream of *HP* [24]. Hp was the first serum protein found to be polymorphic, with two co-dominant alleles, *HP*^1^ and *HP*^2^, giving rise to three distinct common phenotypes: Hp1-1, Hp2-1, and Hp2-2 [25]. DNA sequence analysis suggested that *HP*^2^ is generated by a 1.7 kb intragenic DNA duplication of a tandem two-exon (exons 3 and 4) segment of *HP*^1^ (Figure 1) [24,26,27,28]. Hp is composed of an α chain and a β chain, Hp 1 is a tetramer of two α chains and two β chains, and Hp 2 and Hp 2-1 are multimeric proteins of more than a tetramer [16]. The α chain is encoded by exons 1 to 4 (Hp1) or exons 1 to 6 (Hp2), and the β chain is encoded by exon 5 (Hp1) or exon 7 (Hp2) [24]. It has been reported that Hp2-2 has lower antioxidant activity than Hp1-1 because Hp2-2 binds less efficiently to Hb than Hp1-1 [29]. It is also known that the serum concentration of Hp2-2 is lower than that of Hp1-1 and Hp2-1 [30,31,32].

In addition to the common polymorphisms, two single nucleotide polymorphisms (SNPs), rs5471 and rs2000999, were reported to be associated with the blood concentration of Hp. Rs5471 (-61A > C) in the promoter region of *HP* is characteristic of African populations, and the rs5471 C allele has been reported to be responsible for the Hp2-1 modified phenotype in which fewer Hp2 polypeptides are synthesized than Hp1 polypeptides [33]. Subsequently, it was reported that this variation was correlated with low serum Hp levels [34]. On the other hand, rs2000999 (G > A) in intron 2 of *HPR* was first identified as a genetic variation affecting serum cholesterol levels through a genome-wide association study [35], and the rs2000999 A allele was subsequently reported to be correlated with the low serum Hp levels [36]. The relative positions of rs5471 and rs2000999 are shown in Figure 1.

## 3. Identification of Complete Haptoglobin Gene Deletion, *HP^del^*

In the field of forensic medicine, Hp phenotyping by electrophoresis using starch or polyacrylamide gels was used to determine paternity until DNA testing was introduced in the late 1980s [37]. In some paternity test cases, there were occasionally the so-called single locus exclusion cases in which paternity was excluded based only on the results of Hp phenotyping (anomalous inheritance of Hp). An example is shown in Figure 2. The father was determined to have a Hp2 phenotype because very faint bands equivalent to Hp2 can be seen. His child (child 1) was determined to be Hp1. Normally, this combination is not possible. That is, a parent homozygous for Hp2 cannot have a child homozygous for Hp1. The father and another child (child 2) had low serum haptoglobin levels of 0.12 and 0.006 mg/mL, respectively, indicating hypohaptoglobinemia. Because hypohaptoglobinemic (low serum Hp) and/or anhaptoglobinemic (no serum Hp) individuals are often observed in such families, *HP*^0^, a silent allele of the Hp locus had been suggested to be responsible for such low or no serum Hp [14]. Serum Hp concentrations in individuals with *HP*^1^/*HP^del^* are generally higher than those with *HP*^2^/*HP^del^* for unknown reasons [30]. In addition, Yoshioka et al. screened serum Hp in 9771 Japanese individuals and found one case of anhaptoglobinemia in which serum Hp was not detected even by the highly sensitive enzyme-linked immunosorbent assay (ELISA) method [38]. Southern blot analysis and PCR analysis of the promoter region of HP suggested that this individual was homozygous for complete deletion of the haptoglobin gene (*HP^del^*). This deletion was estimated to extend from the upstream promoter region of *HP* to intron 4 of *HPR* (Figure 1) [14]. Furthermore, as shown in Figure 2, *HP^del^* was found to be in heterozygous state in families with hypohaptoglobinemia, which further explained the anomalous inheritance of Hp [14]. Thus, *HP^del^* was first identified as an *HP*^0^ allele.

## 4. Anaphylactic Shock following Blood Transfusion Due to HP Deficiency

There have been reports of patients suffering from anaphylactic shock due to the production of IgE-type anti-Hp antibodies in addition to IgG-type antibodies after blood transfusion or infusion of plasma components [9]. First, two Hp-deficient patients with anti-Hp antibodies who developed severe anaphylactic shock after infusion of blood products were reported [7,8]. One of them was a woman in her 30s who was 31 weeks into her first pregnancy. She was admitted to the hospital with threatened preterm labor due to chronic polyhydramnios. Lower limb edema with increased amniotic fluid and decreased serum albumin occurred, so a 25% albumin solution was injected. After administering a few drops, she had a severe anaphylactic reaction. The second case was a woman in her 90s with myelodysplastic syndrome. She received three transfusions of red blood cells and one transfusion of platelet concentrate over a 7-month period but had no adverse reaction. However, a month later, when she received a transfusion of platelet concentrates, she suffered an anaphylactic reaction. Both patients’ symptoms improved with corticosteroid treatment. They both had anti-Hp antibodies and no Hp was detected in their sera [7,8].

Genomic DNA from B lymphocytes transformed with Epstein–Barr virus were extracted from these two patients, and they were found to be homozygous for *HP^del^* by southern blot analysis [15]. However, at that time, the exact region of the deletion was not identified and the human genome had not yet been completely sequenced, so diagnosis of *HP^del^* required laborious Southern blot analysis. Therefore, to determine the exact breakpoints of the *HP^del^* allele, cassette-mediated PCR was performed as shown in Figure 3. Genomic DNA of an *HP^del^* homozygote was digested with five restriction enzymes of *Alu*I, *Dra*I, *Eco*RV, *Hae*III, and *Ssp*I, and cassette DNA was ligated to both ends of the DNA fragment. Then cassette-mediated PCR (nested PCR) was performed using cassette primers and HPR exon 5 (the region presents in *HP^del^*) primers. The DNA sequence of the longest PCR product of 2.5 kb, which was obtained from the *Dra*I library, was determined to identify the breakpoints [15]. As shown in Figure 1, the 5′ breakpoint of the deletion was found to be located 5170 bp upstream of the 5′ end of exon 1 of *HP*, and the 3′ breakpoint was located between 52 and 53 bp upstream of exon 5 of *HPR*. Therefore, the size of the deletion is estimated to be about 28 kb. The DNA sequences flanking the 5′ and 3′ breakpoints showed no significant DNA sequence similarity with the junction region of the deletion, except for two bases (TG). Identification of deletion breakpoints has made it possible to perform simple genetic diagnosis of *HP^del^* using methods such as the PCR described below.

In addition to these two cases, several cases of anaphylactic shock due to *HP^del^* in Japanese, Korean, and Chinese patients have been reported [39,40,41,42,43].

## 5. Development of Several Methods for Genetic Diagnosis of *HP^del^*

Several types of diagnostic methods, depending on the site, have been developed so far [44]. Here we will outline these methods. Of course, genetic diagnosis of *HP^del^* can be performed by Southern blot analysis, but it is a laborious and complicated method that is rarely performed at present, so we will not describe it here. Furthermore, although Sanger sequencing of PCR products is the most reliable method for detecting genetic variations, the presence or absence of deletions can be confirmed by the presence or absence of PCR amplification products, so it will not be described here.

### 5.1. Conventional PCR Method for Detection of HP^del^

PCR that spans the deletion junction region can determine the presence or absence of *HP^del^*, but it cannot determine the zygosity of *HP^del^*, that is, whether it is a homozygous (null zygote, *HP^del^/HP^del^*) or heterozygous (hemizygote, *HP/HP^del^*) deletion. On the other hand, exon 1 of *HP* cannot be amplified in *HP^del^/HP^del^* because this region is deleted in *HP^del^* but can be amplified in *HP/HP^del^* and *HP/HP.* Therefore, a duplex conventional PCR method was developed to simultaneously amplify the region spanning the deletion junction (315 bp) and the region flanking exon 1 (476 bp) [15]. Using this method, the presence or absence of the deletion and its zygosity can be determined simultaneously by a single PCR (Figure 4A).

### 5.2. TaqMan Real-Time PCR for Detection of HP^del^

Detection of *HP^del^* by conventional PCR is an adequate method for genetic diagnosis of a small number of specimens. It also has the advantage of low initial cost. However, it is not suitable for genetic diagnosis of large numbers of samples because it takes three to four hours from extraction of genomic DNA to PCR amplification, followed by determination by agarose gel electrophoresis. Based on these limitations, a genetic diagnosis method using TaqMan PCR was developed with the aim of introducing it to relatively large medical institutions, because it eliminates the need for DNA extraction, is simpler and faster, and avoids carry-over of the amplified product because it can be completed in a closed tube [45].

In this method, similar to the conventional PCR method, a region spanning the deletion junction (129 bp) and a region adjacent to exon 1 (84 bp) are simultaneously amplified in a 20 μL reaction system, while using 1 μL of blood diluted 100-fold with 50 mM NaOH as a template instead of purified DNA (Figure 1). As shown in Figure 4B, amplification is detected using TaqMan (hydrolysis) probes labeled with two different fluorescent dyes [46]. Using this method, *HP^del^* zygosity can be diagnosed within 1 h 30 min after blood collection, and although real-time PCR equipment is required, the cost per sample is estimated to be around USD 1. Compared to conventional PCR, this method is a more rapid and inexpensive genetic testing method that can be used on a large number of samples.

### 5.3. SYBR Green I-Based Real-Time PCR Method for Detection of HP^del^

A duplex SYBR Green I-based real-time PCR assay was developed to determine *HP^del^* zygosity in a single tube [47]. Three primers consisting of two forward primers spanning the deletion junction (for *HP^del^*) and *HPR* intron 4 (for *HP*) and a common reverse primer located in *HPR* intron 4 (Figure 1) are used for PCR amplification, and then melting curve analysis is performed. Two distinct melting peaks corresponding to *HP^del^* (amplification size = 124 bp and Tm = 80.3 °C) and *HPR* intron 4, which is deleted in *HP^del^* (amplification size = 148 bp and Tm = 84.5 °C) could be clearly distinguished (Figure 4C).

We can also use 1 μL of blood samples diluted 64- to 1024-fold with 50 mM NaOH solution as a template. The results obtained with this method were in complete agreement with those obtained by the TaqMan-based real-time PCR method. This method is easy to apply compared to TaqMan-based real-time PCR methods because it has a lower initial cost and can be analyzed using economical single color real-time PCR equipment.

### 5.4. Loop-Mediated Isothermal Amplification Reaction Method for Detection of HP^del^

Loop-mediated isothermal amplification (LAMP)-based screening [48] for *HP^del^* was performed using genomic DNA as a template and primer sets optimal for amplification of the spanning region of *HP^del^* and the 5′ region of *HP* in two different tubes [49]. This method also works well using blood samples diluted 100-fold in 50 mM NaOH or blood samples diluted 2-fold in water and then boiled as templates [49]. The advantages of this method are that the reaction is isothermal, and the amplification can be determined by the turbidity of the solution and diluted blood can be used as a template, so no special equipment is required (Figure 4D), and the reaction time is short (about 30 min) [48]. On the other hand, the disadvantages include the requirement of two reaction tubes to determine the zygosity of *HP^del^*, and four to six different primers for amplification in one reaction.

## 6. Immunological Methods for Screening of *HP^del^*

### 6.1. Enzyme-Linked Immunosorbent Assay for Detection of HP^del^

In addition to genetic diagnosis, methods for detecting *HP^del^* include methods for measuring serum Hp, such as measuring peroxidase activity by complex formation of Hp and Hb, simple radioimmunoassay, immunoturbidimetry, and immunonephelometry methods. Among these, the immunoturbidimetry and immunonephelometry methods are easy to perform and are widely used as clinical tests. However, since the detection limit of these methods is several mg/mL, it is difficult to differentiate between hypohaptoglobiemia and anhaptoglobiemia. The ELISA method, which has a detection sensitivity of several μg/mL, can be used as a highly sensitive detection method to compensate for these drawbacks. ELISA using an anti-HP antibody developed by Shimada et al. has a short performance time of 40 min, and is a useful testing method because it can handle a relatively large number of specimens at a time [50].

### 6.2. Latex Agglutination Method for Detection of HP^del^

Recently, a new method for measuring Hp concentration by the latex agglutination method using an automatic analyzer was developed [13]. This method uses a mouse monoclonal antibody conjugated with carboxylate-modified polystyrene latex beads. In this method, a linear absorbance curve was not obtained for the normal haptoglobin range (19–170 mg/dL) but was in the low Hp concentration range and Hp-deficient ranges with a detection limit of 75 μg/dL. To confirm the results, samples with low protein concentrations determined by this method were re-examined by ELISA and *HP^del^* was detected by PCR. Compared to ELISA, the advantages of this method are that it is automatable and inexpensive, which would make it useful for large-scale screening of blood donors. In addition, two anhaptoglobinemic individuals and 21 hypohaptoglobinemic individuals (Hp concentration below 300 μg/dL) were detected using this method in the screening of 7476 samples. The two anhaptoglobinemic individuals were homozygous for *HP^del^*, while 19 of the 21 hypohaptoglobinemic ones were heterozygous for *HP^del^* [13]. Of the two hypohaptoglobinemic individuals without *HP^del^*, one was *HP*^2^/*HP*^2^ and the other was *HP*^1^/*HP*^2^.

## 7. Protein-Based Methods for Detection of Anti-Hp Antibodies in Hp-Deficient Patients

In several case reports, anti-Hp antibodies in Hp-deficient patients were examined. This method differs from the two protein-based detection methods mentioned in the previous two sections in that it detects anti-Hp antibodies rather than Hp itself and, it is an effective method for investigating the cause rather than prevention of anaphylactic shock. In many cases, ELISA and western blotting are used to detect anti-HP antibodies [8,9,12,39,40,41]. Recently, instead of ELISA and western blotting methods, a surface plasmon resonance (SPR) method for detection of anti-Hp antibodies in serum was developed [43]. This method detected IgG type anti-Hp antibodies in the serum of an Hp-deficient Chinese patient living in the United States who developed anaphylactic shock after a blood transfusion. Although SPR appears to be less sensitive than ELISA or western blot methods, its advantages are that fewer biological reagents are required, it is easy to perform, and measurements can be done in real time, resulting in faster results. Therefore, SPR provides a rapid and easily available method for detecting clinically significant anti-HP antibodies, making it a potentially useful test for preventing in addition to investigating the cause of adverse post-transfusion effects such as anaphylactic shock.

## 8. Fluorescent Probe-Based Real-Time PCR Method for Simultaneous Detection of *HP^del^* and Other *HP* Polymorphisms

Two fluorescent probe-based real-time PCR methods for simultaneous detection of *HP^del^* and other *HP* polymorphisms have been developed. One was detection of *HP^del^* and common *HP* polymorphisms by adding *HP*^2^-specific TaqMan probe and primers to *HP^del^* TaqMan probe and primers and *HP5*′ TaqMan probe and primers [51]. Copy numbers of *HP*^2^ (0, 1, 2) were determined by ΔΔCt method using copy numbers of *HP5*′ as a control and whether or not there was an *HP^del^* by endpoint genotyping assays [51,52]. The other was the simultaneous detection of *HP^del^* and rs2000999 G > A variation by endpoint genotyping assays and fluorescence melting curve analysis (FMCA) [53]. With this method, the rs2000999 G > A variation was detected by FMCA, one of the most robust SNPs detection methods [54]. Unlike the TaqMan assay, no degradation of the fluorescent probe in the FMCA would be preferable, but both *Taq* polymerases with and without 5′-3′ exonuclease activity have been reported to be available [54].

These two methods are useful not only for screening *HP^del^* but also for association studies of these *HP* polymorphisms on a relatively large scale, especially in East and Southeast Asian populations.

## 9. Geographic Distribution of *HP^del^*

The distribution of *HP^del^* in various populations was determined using genetic diagnostic methods. As shown in Table 1, *HP^del^* has so far been detected in East Asian (Japanese, Korean, Chinese, Taiwanese, Mongolian) and Southeast Asian (Thai, Indonesian, Vietnamese) populations with a frequency of approximately 0.9% to 4%, and therefore the frequency of *HP^del^* homozygotes was estimated to be approximately 1 in 640 to 12,600 people [15,30,32,45,55,56,57,58]. On the other hand, it has not been found in Tibetans, Africans, Europeans, West Asians, or people from the Americas [15,30,51,59,60]. Notably, *HP^del^* has not been detected in Africa, where anhaptoglobinemia was first reported 65 years ago [61]. Although the rs5471 C allele is thought to be significantly associated with low serum Hp concentrations [34], no genetic variant causing anhaptoglobinemia has yet been found in African populations. Therefore, especially in malaria-endemic areas in Africa, many cases of anhaptoglobinemia are thought to be acquired secondarily, probably due to malaria-induced hemolysis or the like [62].

On the other hand, among populations in East Asia and Southeast Asia, the highest frequency of *HP^del^* is observed in the Chinese population, so it is assumed that this genetic variation originated in China and spread to surrounding regions. Surprisingly, *HP^del^* has not been observed in Tibetan populations, where a large-scale gene flow from lowland southern China is thought to have occurred [63]. Although the reason for this is not certain and requires further detailed analysis of East and Southeast Asian populations, it may suggest that *HP^del^* arose relatively recently [60].

## 10. *HP*-Deficient Alleles Other Than *HP^del^*

In addition to *HP^del^*, a gene variation in the splice donor site of the *HP* that is specific to Irish people has been reported as a causative gene for HP deficiency [64]. This variation (NM_001126102.1:c.190 + 1 G > C) changes the first two bases of intron 3 of *HP* from GT to CT, which prevents normal splicing and causes nonsense mutations to appear at an early stage (Figure 1). It is thought that not only abnormal mRNA quality but also nonsense-mediated mRNA degradation occurs. This is a genetic variation that is presumed to result in an extremely low expression of *HP* mRNA, resulting in Hp deficiency. Data from whole-genome sequencing of 8453 Irish individuals gave an estimated frequency of this variation of 0.56%. Furthermore, in an analysis of 150,656 people, six people who were homozygous for this variation were detected, five of whom lived to between 47 and 69 years of age, and one person who lived until 91 years of age. Therefore, as also shown in the case of *HP^del^*, it was further suggested that Hp is not essential for human survival, or at least does not shorten the lifespan, while it has the physiologically important function of removing harmful free Hb. However, it is estimated that such individuals, like homozygotes of *HP^de^*^l^, are at risk of developing severe adverse reactions after blood transfusion due to anti-Hp antibodies.

## 11. Conclusions

Many of the causes of posttransfusion anaphylaxis are unknown, but among those caused by serum protein deficiency, Hp deficiency is the most common cause in Japan and probably in East and Southeast Asians [10,11]. *HP*^del^ is a genetic variation discovered by chance through analysis of cases of discrepancies in parent–child relationships in forensic medicine practice. It is also thought to have played a role in determining the existence of congenital HP defects, which had long been questioned by some researchers [62,65]. Also, considering the variations in various public databases, there are probably no alleles that have a similar frequency to the *HP^del^* in other human populations.

*HP^del^* is a germline variation, and serious adverse blood transfusion reactions due to this variation can be prevented by performing genetic testing once in a lifetime, at the low cost of about 1 USD per sample. Conventional PCR and detection of anti-Hp antibodies are considered sufficient to determine whether adverse reactions caused by administration of blood transfusion products are due to Hp deficiency caused by *HP^del^*. However, for screening homozygotes of *HP^del^* with a high potential risk of adverse reactions after blood transfusion, the TaqMan method and SYBR green I method using real-time PCR, which can test a large number of samples, are more useful. In addition, among protein-based methods, the latex agglutination method using an autoanalyzer may be the most suitable for screening large numbers of samples. Similarly, in order to administer plasma products to such patients or establish a donor pool for Hp-deficient individuals, it may be desirable to use real-time PCR-based tests to identify homozygotes for *HP^del^*.

## Figures and Tables

**Figure 1 biomedicines-12-00790-f001:**
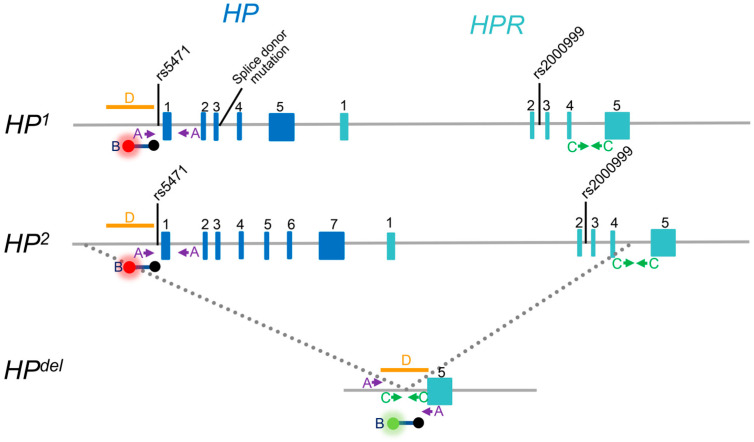
Gene structures of HP (*HP1*, *HP2*, and *HP^del^*) and HPR. The exons of *HP* and *HPR* are indicated by blue and light blue boxes, respectively. The relative positions of rs5471, rs2000999, and a splice donor mutation are indicated. In addition, approximate locations of PCR primers used for conventional PCR (A), SYBR-green I-based real-time PCR (C), and LAMP (D) and probes for TaqMan real-time PCR (B) for detection of *HP^del^* are indicated.

**Figure 2 biomedicines-12-00790-f002:**
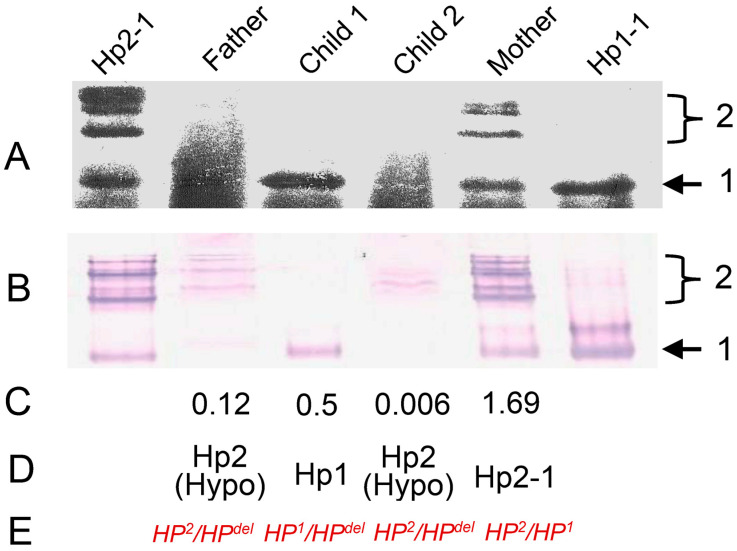
A family with anomalous inheritance of Hp that led to the identification of *HP^del^*. (**A**) The Hp phenotype was determined by mixing 20 μL serum with 10 μL hemolysate, followed by electrophoresis in a 5% polyacrylamide gel. The gel was immersed in a 40% acetic acid solution containing 0.4% leucomalachite green for 15 min and then was stained with 3% hydrogen peroxide. (**B**) Western blotting using anti-Hp antibody. (**C**) Serum Hp levels determined by ELISA. (**D**) Resultant phenotypes. (**E**) The genotypes deduced by semiquantitative Southern blotting. The numbers 1 and 2 on the right indicate bands derived from Hp1 and Hp2 polypeptides, respectively.

**Figure 3 biomedicines-12-00790-f003:**
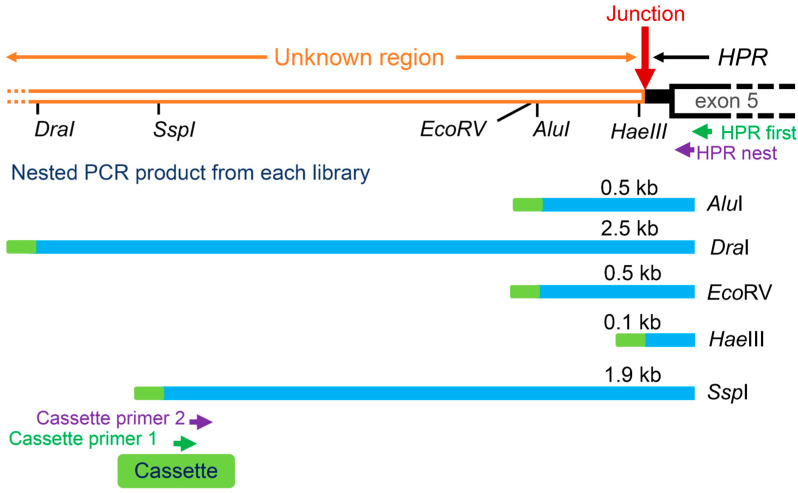
Strategy for cloning deletion break points of *HP^del^*. Restriction sites near the junction of *HP^del^*, fragments obtained from each enzyme library, and relative positions of primers used are indicated. DNA of a homozygote of *HP^del^* was digested by five restriction enzymes, ligated with a cassette, and libraries were obtained. Nested PCR was performed using two sets of HPR-specific and cassette primers. The sequence of the longest product of 2.5 kb, obtained from the *Dra*I library, was determined.

**Figure 4 biomedicines-12-00790-f004:**
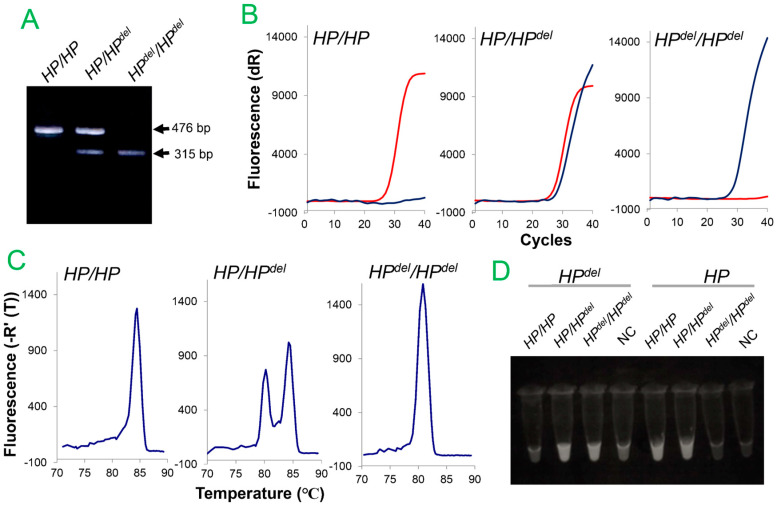
Representative results obtained using each test method for detection of *HP^del^*. The results of conventional PCR (**A**), TaqMan real-time PCR using CAL Fluor Red 610-labeled probe to detect the 5′ region of *HP* and FAM-labeled probe to detect *HP^del^*. The red line is the CAL Fluor Red 610 (*HP*) signal and the blue line is the FAM (*HP^del^*) signal. (**B**), SYBR-green I-based real-time PCR (**C**), and LAMP (**D**) are shown. Purified DNA or diluted blood samples of individuals without (*HP*/*HP*), heterozygote (*HP*/*HP^del^*), or homozygote (*HP^del^*/*HP^del^*) of *HP^del^* were used as templates. NC stands for negative control.

**Table 1 biomedicines-12-00790-t001:** Frequency of *HP^del^* in various populations.

Populations	Total Number of Alleles	*HP^del^* Frequency (%)	Estimated Homozygotes	References
East Asia				
Mongolians	2130	0.9	1/12,600	[58]
Han Chinese	11,942	4.0	1/640	[15,32]
Koreans	1332	2.9	1/1200	[15,55]
Japanese	12,404	1.7	1/3500	[15,30]
Taiwanase	1962	2.9	1/1200	[57]
Central Asia				
Tibetans	364	0	0	[58]
Tamangs	106	0	0	[58]
Uygurs	112	0	0	[58]
Southeast Asia				
Indonesians	210	1.0	1/11,000	[45]
Thais	400	1.5	1/4400	[56]
Vietnamese	588	2.0	1/2400	[58]
South Asia				
Bangladeshi	102	0	0	[58]
Tamils	104	0	0	[58]
Sinhalese	102	0	0	[58]
West Asia				
Turks	214	0	0	[58]
Europe				
Europeans	400	0	0	[15,30]
Africa				
Xhosans	202	0	0	[15]
Ghanaians	246	0	0	[51]
Gambians	1196	0	0	[59]
America				
Mexicans	372	0	0	[60]
Puerto Ricans	160	0	0	[60]
Colombians	140	0	0	[60]
Peruvians	140	0	0	[60]

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
