# Peer review of "Identification and Diagnosis of Complete Haptoglobin Gene Deletion, One of the Genes Responsible for Adverse Posttransfusion Reactions"

_biomedicines, 2024, doi:10.3390/biomedicines12040790_

Round 1
Reviewer 1 Report
Comments and Suggestions for Authors
The work entitled “Identification and diagnosis of complete haptoglobin gene deletion, the gene responsible for adverse posttransfusion reactions” reviews the importance of defining the type of haptoglobin (Hp) allele present in the person who will receive a blood transfusion or platelets. It clearly establishes the effect of the presence of the null allele, in which case people can develop antibodies against Hp and, therefore, develop them in a reduced number of cases in the case of transfusion.
The authors describe the presence of two co-dominant alleles, Hp1 and Hp2, in addition to Hp-null and two polymorphisms, rs5471 and rs2000999, located in the vicinity of the Hp promoter and another in that of HPR, respectively.
Although it reviews the elements involved in the anaphylactic response and its relationship with the Hb allele, the work delves into the detection methods of allelic variants to avoid iatrogenesis in the blood recipient. Conventional PCR methods are proposed, as are the use of TaqMan probes, real-time amplifications with SYBR-Green, isothermal LAMP amplification processes, immunological tests of immunoadsorbent assays or simply ELISA tests, and latex agglutination. Finally, they describe the allelic distribution in various geographical points.
The article is well written, adequately compiles the two effects of allelic variants on the blood or platelet recipient, and sufficiently illustrates the relevance of early detection of these haplotypes, particularly in the East and Southeast Asian populations.
I recommend its acceptance in the form in which the review is found.
Author Response
To Reviewer #1
Thank you very much for your evaluation of our manuscript.
Reviewer 2 Report
Comments and Suggestions for Authors
The review of Mikiko Soejima and Yoshiro Koda contains some interesting elements. Although the general interest for clinicians and transfusion medicine practitioners there are improvements that are necessary before the article can be accepted.
English is very poor, I suggest to correct the manuscript with the help of an editing service or a mother tongue writer, since, unfortunately, some arguments result unclear.
Figure 2 is confused, it is not clear which is the inheritance, whata are the arrows and number on the right and agarose was confused with western blot. This figure has to be much more clear to explaine clearly the case.
In Table 1 what is the ‘’number of chromosomes’’???
In my opinion it is necessary to add a chapter on sequencing techniques, in particular sanger sequencing is cheap and routinely available in most of the hospitals nowadays.
Finally, regarding the pattern of inheritance I think that its is necessary to further discuss it, this is quite unclear to me, it seems that HP1/HPdel and HP2/HPdel have different phenotypes, since from the fig 2 HP1/HPdel has higher serum protein than HP2/HPdel. This effect can be due to allelic enhancement? If its is the case please refer to
https://onlinelibrary.wiley.com/doi/pdf/10.1111/j.1365-3148.2005.00603.x https://www.sciencedirect.com/science/article/pii/S1473050222003287
and explain the case, otherwise please detail the genetic background of these muations/deletions.
Comments on the Quality of English Language
English is very poor and need extensive revision
Author Response
To Reviewer #2
Thank you very much for your evaluation of our manuscript.
Following your suggestions, we revised our manuscript. Change was made as follows:
Q1: English is very poor, I suggest to correct the manuscript with the help of an editing service or a mother tongue writer, since, unfortunately, some arguments result unclear.
Our reply: Thank you for your valuable suggestion. According to your suggestion, we requested editing service to improve our English. I hope it has improved.
Q2: Figure 2 is confused, it is not clear which is the inheritance, whata are the arrows and number on the right and agarose was confused with western blot. This figure has to be much more clear to explain clearly the case.
Our reply: Thank you for your valuable comment. According to your suggestion, we added a sentence about the inheritance of the case in the text “That is, a parent homozygous for Hp2 cannot have a child homozygous for Hp1.” (p. 3, lines 104-105). In addition, we modified the Figure 2 and its legend by adding explanations including the number on the right, acrylamide gel electrophoresis, and Western blotting (p. 4, lines 121-127).
Q3: In Table 1 what is the ‘’number of chromosomes’’???
Our reply: Thank you for your pointing it out. Following to your point, we changed from “number of chromosome” to “2 × number of subjects”.
Q4: In my opinion it is necessary to add a chapter on sequencing techniques, in particular sanger sequencing is cheap and routinely available in most of the hospitals nowadays.
Our reply: Thank you for your helpful comment. We added the sequencing techniques to the chapter of Southern blotting as “Furthermore, although Sanger sequencing of PCR products is the most reliable method for detecting genetic variations, the presence or absence of deletions can be confirmed by the presence or absence of PCR amplification products, so this will not be described here.” (p. 5, lines 176-179).
Q5: Finally, regarding the pattern of inheritance I think that its is necessary to further discuss it, this is quite unclear to me, it seems that HP1/HPdel and HP2/HPdel have different phenotypes, since from the fig 2 HP1/HPdel has higher serum protein than HP2/HPdel. This effect can be due to allelic enhancement? If its is the case please refer to https://onlinelibrary.wiley.com/doi/pdf/10.1111/j.1365-3148.2005.00603.x https://www.sciencedirect.com/science/article/pii/S1473050222003287 and explain the case, otherwise please detail the genetic background of these muations/deletions.
Our reply: Thank you for your valuable comment. As you mentioned, child 1(HP1/HPdel) has higher serum protein than father or child 2 (HP2/HPdel) in this family. However, the cause of this phenomenon is unknown. Therefore, we added the following sentence: “Serum Hp concentrations in individuals with HP1/HPdel are generally higher than those with HP2/HPdel for unknown reasons.” (p. 3, lines 110-111).
Reviewer 3 Report
Comments and Suggestions for Authors
Overall, the manuscript is well written and concerns an important topic in biomedicine. There are some minor points that need to be modified before publication:
-The title of the manuscript is disturbing. You get the impression that only Hp deletions are responsible for adverse posttransfusion reactions. Shhould read "one of the genes responsible..."
- The authors mention in the introduction that washing of RBCs can be helpful. More details are needed here.
-Figure 2 is not fully relevant. The legend should at least be extended to facilitate the reading.
- The number of self-citations is too high. The authors have excluded some of the more important papers in the field. A more careful choice of references is needed.
Author Response
To Reviewer #3
Thank you very much for your evaluation of our manuscript.
Following your suggestions, we revised our manuscript. Change was made as follows:
Q1: The title of the manuscript is disturbing. You get the impression that only Hp deletions are responsible for adverse posttransfusion reactions. Should read "one of the genes responsible..."
Our reply: Thank you for your valuable suggestion. According to your suggestion, we changed the title (p. 1, line 3).
Q2: The authors mention in the introduction that washing of RBCs can be helpful. More details are needed here.
Our reply: Thank you for your valuable comment. According to your comment, we changed the description about the washing of RBCs (p. 2, lines 47).
Q3: Figure 2 is not fully relevant. The legend should at least be extended to facilitate the reading.
Our reply: Thank you for your pointing it out. Following to your point, we modified the Figure 2 and its legend (p. 4, lines 121-127). I hope it has improved to facilitate the reading.
Q4: The number of self-citations is too high. The authors have excluded some of the more important papers in the field. A more careful choice of references is needed.
Our reply: Thank you for your valuable suggestion. According to your suggestion, we deleted two our own papers (originally, 57 and 58) and included two references (37 and 44).
Round 2
Reviewer 2 Report
Comments and Suggestions for Authors
As I stated in the previous revision english should be improved, in particular in paragraph 3
Comments on the Quality of English LanguageAs I stated in the previous revision english should be improved, in particular in paragraph 3
Author Response
To Reviewer #2
Thank you very much for your evaluation of our manuscript.
Following your suggestions, we revised our manuscript to correct the English with the help of a colleague whose native language is English.